

# Frequency-Dependent Shear Wave Attenuation across the Central Anatolia Region, Türkiye

Gizem Izgi[1], Tuna Eken[2], Peter Gaebler[3], Tülay Kaya-Eken[4], and Tuncay Taymaz[2]

[1]University of Potsdam Institute of Geosciences Karl-Liebknecht-Str. 24-25, *14476* Potsdam-Golm, Germany
[2]Istanbul Technical University, Department of Geophysical Engineering, Maslak, Sarıyer, *TR-34467, Istanbul*, Türkiye
[3]Federal Institute for Geosciences and Natural Resources (BGR), Stilleweg 2, *30655* Hannover, Germany
[4]Boğaziçi University, Kandilli Observatory And Earthquake Research Institute, *34684* Çengelköy-İstanbul/Türkiye

**Correspondence:** Gizem Izgi (gizem.izgi@uni-potsdam.de)

**Abstract.** The Central Anatolian Plateau with its volcanic provinces represents a broad transition zone between the compressional deformation in the east and the extensional regime in the west. The Central Anatolian Fault Zone separates the Kırşehir Block in the north and the Anatolide-Tauride block in the south within the plateau. A proper understanding of physical properties such as seismic attenuation in the crustal volume of this region can provide hints toward the possible source for the geodynamic events in the past and present that likely leads to the observed deformation. To model intrinsic and scattering attenuation separately, we perform a non-empirical coda wave modeling approach in which a fitting process between observed and synthetic coda wave envelopes is performed for each earthquake in multiple frequency bands. Here acoustic radiative transfer theory assuming multiple isotropic scattering was utilized for the forward modeling of the synthetic coda-wave envelopes of local earthquakes. Our findings generally highlight the prominent nature of intrinsic attenuation over scattering attenuation implying the presence of thick volcanic rocks with relatively high attenuation values beneath Central Anatolia. In overall the spatial distribution of the attenuation at varying frequencies marks the Kırşehir Massif distinctively with its considerable high attenuating character. Our findings together with early seismological and geo-electrical models suggest a possible partial melt beneath the most of Central Anatolian Volcanic Province and resultant zones of elevated fluid rich content exhibit high and dominant intrinsic attenuation. To the southeast, a gradual decrease in the observed attenuation coincides with the Central Tauride Mountains where high altitude is considered to be evolved following the slab break-off and resulting mantle upwelling.

## 1 Introduction

As being overly complex with a long history of subduction, collision, and accretion, acting on the Anatolian Plate, the Central Anatolian Fault Zone (CAFZ) has been widely studied (Koçyiğit and Beyhan, 1998; Okay and Tüysüz, 1999). An identification of crustal strain accumulation and its release in time in relation to the spatio-temporal variations of crustal deformation in this tectonically complicated area is a hard task (McKenzie, 1972; Jackson and McKenzie, 1984; Şengör and Yilmaz, 1981). The study of seismic attenuation provides valuable insights into ongoing magmatic processes within subduction zones. This is because environments with higher temperatures or the presence of fluids can have distinct effects on seismic attenuation compared to seismic velocity (Karato et al., 2003). A proper quantification of intrinsic and scattering attenuation separately in





the region is important for shedding light into the crustal scale deformations expected in the future, which is likely controlled

by the type of depth-varying heterogeneities along the CAFZ and its vicinity. The energy characteristics of seismic waves

decay due to their anelastic properties of the medium in which they propagate. This energy loss can be attributed to the

three major phenomena namely geometrical spreading, intrinsic and scattering attenuation that are critical physical parameters

when considering realistic seismic wave propagation (Aki and Chouet, 1975; Aki, 1969). In the case of intrinsic attenuation

Equation(1a) seismic wave energy loss occurs through transformation into other forms (e.g. heat) and can be caused by friction

or mineral dislocations. Scattering attenuation (Equation1b) on the other hand, can be described by the energy redistribution

due to small-scale heterogeneities along the path.

$$(a) \qquad Q_i^{-1} = \frac{b}{2\pi f} \qquad ,(b) \qquad Q_{sc}^{-1} = \frac{g^* V_0}{2\pi f} \qquad (1)$$

where $Q_i$ and $Q_{sc}$ are intrinsic and scattering attenuation, b is intrinsic absorption parameter, f is the frequency and $g*$ is

scattering coefficient, respectively. Separating scattering and intrinsic attenuation in varying frequency ranges can be achieved

through several methods (Sato et al., 2012). One approach employs a coda wave modeling procedure where forward part of

the problem is achieved by completely analytic expression of synthetic coda wave envelopes based on the radiative transfer

theory (RTT) (Sens-Schönfelder and Wegler, 2006a). RTT was used to investigate source and attenuation in different geological

settings in a seismically active Vogtland region in Germany-Czechia border (Eulenfeld and Wegler, 2016), with a larger data

set over the entire United States (Eulenfeld and Wegler, 2017). In Anatolia, Gaebler et al. (2019) and later Izgi et al. (2020)

employed this approach to investigate the central and northwestern section of the North Anatolian Fault Zone, respectively. In

the present work, we utilize the same RTT approach to further understand the upper crustal part the of Central Anatolia region

by mapping the 2-D variations of frequency-dependent attenuation properties. Our findings will provide new constraints on

the seismic character of the upper crustal part of the study region, for instance, intrinsic and scattering attenuation properties

associated with various interesting tectonic and geological features including Central Anatolian Volcanic Province (CAVP),

that would be meaningful for a proper interpretation of future geodynamic hypotheses aiming at explaining past and/or present

deformation history and tectonic evolution of the study area.

## 1.1 Study Area

Türkiye, as a part of the tectonically active system, has been undergoing constant stress driven by convergent motion of the

Arabian Plate with respect to the Eurasian Plate and the northward convergence of the African Plate beneath Anatolia (Armijo

et al., 1999; Faccenna et al., 2006; Reilinger et al., 2006; Taymaz et al., 1991c; Confal et al., 2018). As a result of this ongoing

Arabian collision and slab rollback along the Hellenic trench, the Anatolian micro-plate has a westward escaping motion with

respect to Eurasia with increasing velocities of $\sim$ 18 to 25 mm/y from east to west (Reilinger et al., 2006). Anatolian micro-

plate's extrusion is accommodated by the North and East Anatolian Fault Zones (NAFZ and EAFZ respectively) as evidenced

by destructive earthquakes (Taymaz et al., 1990, 1991b, a; Şengör and Yilmaz, 1981). The active tectonics of Anatolian micro-

plate is shown in Fig. 1. Apart from these major fault zones, the sinistral CAFZ is one another significant structure that





characterizes internal deformation in Central Anatolia and is considered to form the future eastern boundary of the Anatolian micro-plate following a replacement with the EAFZ (Koçyiğit and Beyhan, 1998; Taymaz et al., 1991a; Melgar et al., 2020; Taymaz et al., 2021). Central Anatolia region (Fig. 1) plays a key role as a transition zone between the compressional regime in the east and the extensional regime in the west. It is characterized by an increasing elevation from the interior to the north

and the presence of the Taurus Mountains to the south (Fig. 2). In this form, the region can be considered to exhibit a typical plateau-like morphology on a smaller scale. The continental fragments were consolidated during the Neo-Tethyan Ocean closure in the Cretaceous. As a result of this closure, the northern Pontides became disconnected from the Kırşehir Block, in the southern region, it was separated from the Anatolide-Tauride block, which is identified by the presence of the Inner Tauride suture (ITS) (Abgarmi et al., 2017; Şengör and Yilmaz, 1981). The Anatolide-Tauride block primarily comprises non-

metamorphosed platform carbonates while Kırşehir Block is composed of Cretaceous high-temperature metamorphic rocks and igneous intrusions (Tüysüz, 1999; Whitney and Hamilton, 2004; Okay and Tüysüz, 1999).

### 1.2 Data

We utilized 72 broadband stations deployed within the framework of a passive seismic experiment between 2013 and 2015 and named as the Continental Dynamics-Central Anatolian Tectonics (CD-CAT) (Portner et al., 2018). In total, we examined

1509 local earthquakes with local magnitudes between $M_l$ 2.0 and $M_l$ 4.6 from the IRIS Data Management Center. The distance between stations and events are restricted to a maximum of 120 km to avoid the effects of Moho-guided Sn-waves. We preferred to analyze the earthquakes with less than 10 km of a focal depth in order to exclude the effect of relatively large-scale heterogeneities on coda wave trains. Station-event pairs are shown in Figure 3.

We used a Butterworth band-pass filter with central frequencies at 0.75, 1.5, 3.0, 6.0 and 12 Hz. To include all direct S-wave

energy we set up a S-wave window in between 3 and 7 s prior to and after S-wave onset, respectively. Theoretical S-wave onsets are determined according to a pre-calculated crustal model (Delph et al., 2015). The coda window ends 100 s after S-onset for each earthquake. Waveforms of all selected earthquakes pose minimum 10 s of coda window and a signal-to-noise ratio greater than 2.5 to ensure the quality of the S-wave coda. A total of 916 local earthquakes, whose spatial distribution is presented in Figure 4, meet these selective criteria.

## 2 Methodology

### 2.1 Inversion Process

Chandrasekhar (2013) introduced the RTT as a means to describe the propagation of light through a turbulent atmosphere. In the field of seismology, RTT has been used to describe the propagation of seismic waves in heterogeneous media (Przybilla and Korn, 2008; Sens-Schönfelder and Wegler, 2006a; Gaebler et al., 2015; Izgi et al., 2020).We utilized an inversion method based

on acoustic RTT introduced by Sens-Schönfelder and Wegler (2006b) since it enables us to use events with smaller amplitudes and provides directly the relation between source excitation and coda wave amplitude. With this method, coda normalization is





**Figure 1.** Active tectonics of Anatolian micro-plate with respect to Eurasian, Arabian and African Plates. Extensional boundaries are shown in green and major shear zones are shown in red (NAF: North Anatolian Fault, EAF: East Anatolian Fault, DSF: Dead Sea Fault, EAAC: East Anatolia Accretionary Complex, CAVP: Central Anatolian Volcanic Province. The study area is indicated by the blue rectangle. Bathymetry and topography data are taken from GEBCO (2019), and active faults are from Şaroğlu et al. (1992).



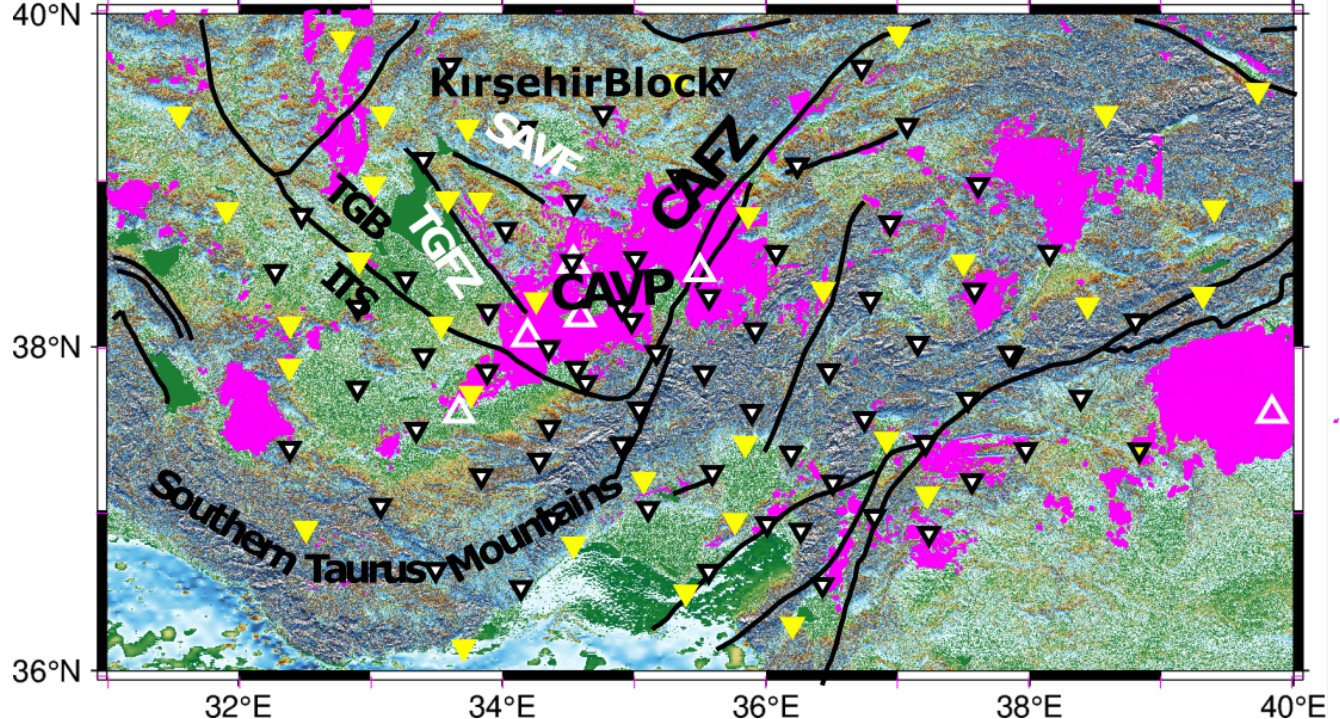

**Figure 2.** The topographic map showing the study area with key tectonic features modified from (Abgarmi et al., 2017). Black lines indicates main faults in the study area (CAFZ: Central Anatolian Fault Zone, ITS: Inner Tauride Suture, SAVF: Savcılı Fault, TGFZ: Tuz Gölü Fault Zone). White inverted triangles indicate Continental Dynamics–Central Anatolian Tectonics (CD-CAT) network and yellow inverted triangles are Kandilli Observatory and Earthquake Research Institute (KOERI) stations. Large white triangles show Holocene volcanoes and magenta polygons are Neogene–recent volcanic deposits (e.g. CAVP: Central Anatolian Volcanic Province).

not compulsory because the source and site parameters are calculated within the inversion. For each earthquake, the inversion simultaneously solves for the intrinsic absorption parameter (b), scattering coefficient ($g*$), site amplification factor ($R_i$) and spectral source energy (W). For the corresponding frequency band of an earthquake, the event must be recorded at minimum three stations otherwise its removed from the analysis. Theoretical envelopes ($E_{mod}$) are calculated via solving the Equation(2) for each Green's function with distance vectors from the station to the hypo-center ($\overrightarrow{r_i}$) for each station with shown with an index *I*. The Green's function of the analytic approximation of the solution for 3D radiative transfer (Paasschens, 1997) is:

$$G(t,r,g_0) = e^{-v_0 t g_0}\left[\frac{\delta(r-V_0 t)}{4\pi r^2} + \left(\frac{4\pi V_0}{3g_0}\right)^{-3/2} \times t^{-3/2}\right.$$

$$\left. \times \left(1-\frac{r^2}{V_0^2 t^2}\right)^{1/8} \times K\left(V_0 t g_0\left(1-\frac{r^2}{V_0^2 t^2}\right)^{3/4}\right) \times H(V_0 t - r)\right] \quad (2)$$



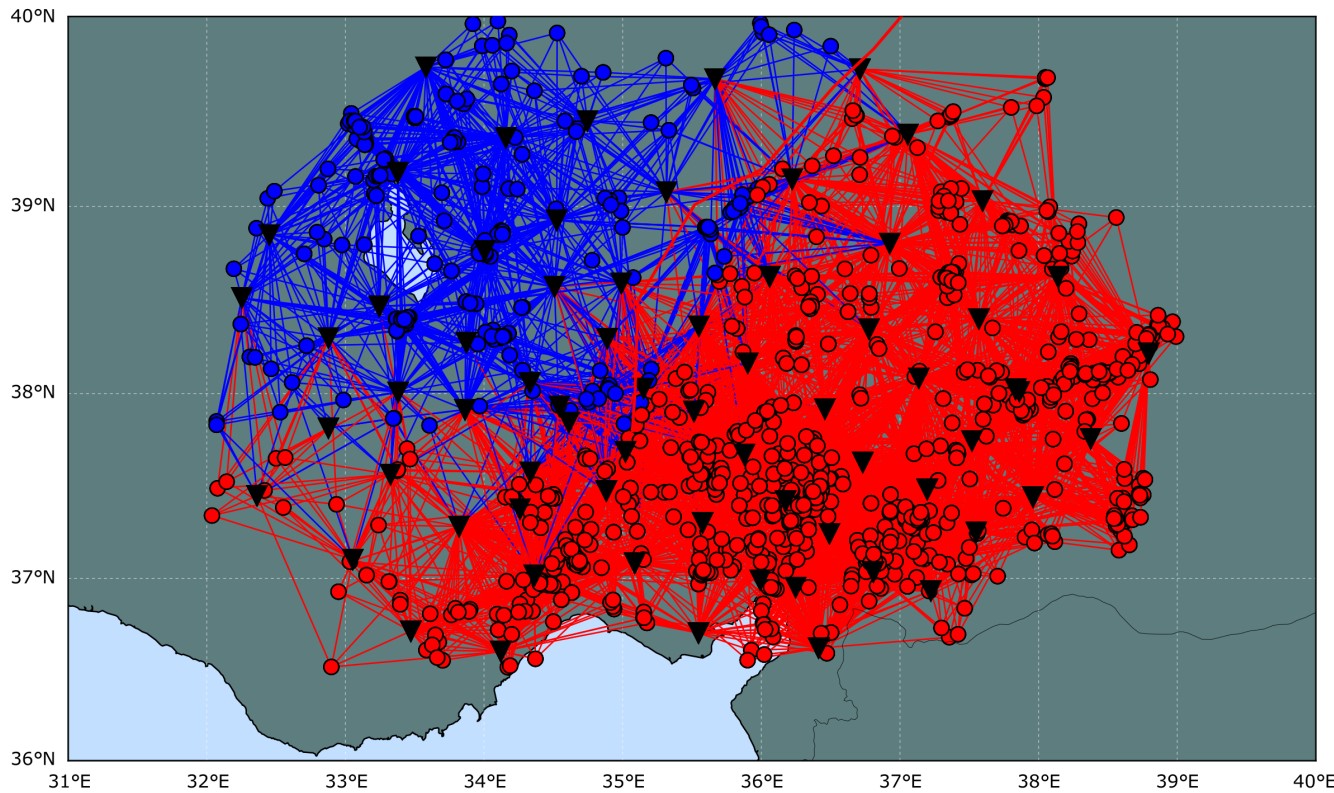

**Figure 3.** 1509 earthquakes given as circles with ray paths. Their colors are shown according to their position relative to the Central Anatolian Fault (CAF) and Inner Tauride Suture (ITS) marks the tectonic difference. Reversed black triangles show the location of 72 broadband stations.

where the term with the Dirac delta function $\delta$ represents the direct wave and the other terms without the $\delta$ is the scattered waves. $V_0$ corresponds to the mean S-wave velocity and $g_0$ is the only scattering parameter in this formula used under the assumption of isotropic scattering. Although this assumption is not realistic, the scattering coefficient is used to determine the transport scattering coefficient $g*$ for more realistic anisotropic scattering (Gaebler et al., 2015). The modeled energy envelopes $E_{mod}$ are compared with the observed energy envelopes $E_{obs}$ for each seismic station within our inversion scheme which is

summarized below. By minimizing the error function between modeled and observed energy densities, optimal intrinsic and scattering attenuation parameters are determined.

$$E_{mod}(t,\overrightarrow{r_i}) = WR_iG(t,\overrightarrow{r_i}g*)e^{-bt} \tag{3}$$

The observed energy densities are calculated by following as a combination of kinetic and potential energy:

$$E_{obs}(t,r) = \rho_0\dot{u}(t,r)^2/C\delta f \tag{4}$$



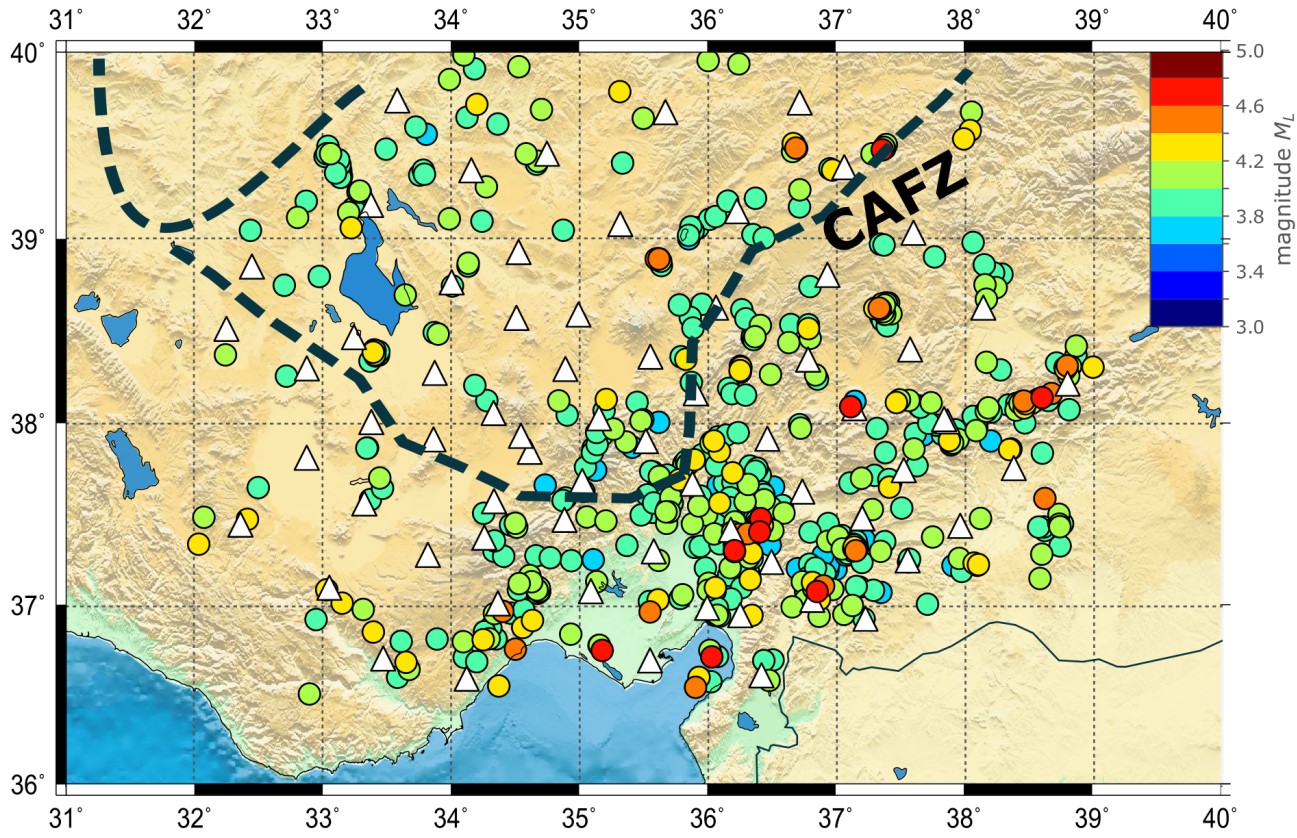

**Figure 4.** Selected 916 earthquakes with varying magnitudes are shown in colors accordingly. White triangles represent 72 broadband stations.

where $\rho_0$ is the mean mass density, $\dot{u}(t,r)^2$ is the mean square velocity and C acts as the free surface correction as C=4 (Emoto et al., 2010). To obtain observed envelopes, the seismic velocity data are filtered by a specific frequency band $f$ and normalized by the filter width $\delta f$. This problem is solved, for each station $(N_s)$ with an index (i), each event $(N_E)$ with an index (j) and for each observed energy densities of them (Nij) in time samples index (k), by minimizing an error function:

$$Err(g) = \sum_{i,j,k}^{N_s,N_E,N_{ij}} (lnE_{obsijk} - lnE_{modijk}(g))^2 \qquad (5)$$

Then for each event the optimized g is obtained to solve the Equation 6 and b, $R_i$ and $W_j$ are obtained.

$$E_{obsijk} = lnG(t_{ijk}, r_{i,j}, g) + lnR_i + lnW_j - bt_{ijk} \qquad (6)$$

For further information on inversion processes, the reader is referred to Eulenfeld and Wegler (2016); Sens-Schönfelder and Wegler (2006a); Izgi et al. (2020); Eken (2019).





## 2.2 Envelope Fitting

The method has been validated by fitting synthetic and observed energy envelopes, utilizing an analytical approximation to calculate direct S- and coda wave energy (Paasschens, 1997). Earlier it has been proven that using only the S-wave energy for fitting the envelopes would be sufficient, as the S-wave dominates the wave-train (Gaebler et al., 2015). Figure 5 presents an example of envelope fitting performed on an event with $M_L = 4.1$ using five different frequency bands (0.75, 1.5, 3.0, 6.0, and 12.0 Hz) at four stations and demonstrates that observed energy densities fit accurately allowing for the precise determination

of direct S-wave onsets in terms of envelope amplitude.

Across all magnitudes and frequency bands, the direct onset of the S-wave can be accurately modeled in terms of envelope amplitude. Furthermore, the decay of the seismic coda can be modeled with high accuracy for time windows of up to 100 s from 27 to 85 km depth while for 105 km depth, the time window resulting in accurate envelope fitting is maximum 50 s. It is worth noting that the seismic coda decays faster at higher frequencies thus lowering the accuracy of the modeled

energy envelopes. Mainly because of geometrical spreading, it is expected to observe a smaller window of the fitting between synthetic and observed energy densities. Thus, the bottom row in Fig. 5 representing the distance equal to 105 km, shows a smaller fit as a function of time and energy densities. Since the top row in Fig. 5 corresponds to the uppermost crustal layer when compared to the others, it shows less attenuative energy density envelopes and precise fitting. The overall quality of the fits is further comparable to those estimated for different tectonic settings in previous studies (Izgi et al., 2020; Gaebler et al.,

2015; Eulenfeld and Wegler, 2016).

## 3 Results

In the study area, the values of all attenuation types increase with decreasing frequency; such as, the estimated total attenuation of seismic waves $(Q_T^{-1})$ ranges from 0.0013 at 12.0 Hz to 0.0216 at 0.75 Hz. The inverse of the scattering quality factor $(Q_s^{-1})$ varies between 0.0003 at 12.0 Hz and 0.0116 at 0.75 Hz. The inverse of the intrinsic quality factor decreases from 0.0110 at

0.75 Hz to 0.0011 at 12 Hz. Intrinsic attenuation tends to show an overall dominancy over scattering attenuation. The lateral variation of attenuation values marks the CAF and ITS. In the southeastern part, lower attenuation values with a gradually decreasing pattern are notable from the CAF towards the southeastern end. The dominance of different attenuation types at different frequency bands, but their overall contribution to total attenuation appears to be similar or close to each other. 2D spatial variation of intrinsic, scattering and total attenuation parameters estimated for five frequency bands are shown in Fig.

6. We present a comparison of frequency-dependent attenuation parameters for the northern and southern part of the CAFZ in Fig. 7. Along the northern part of CAFZ, intrinsic attenuation decreases from 0.0075 at 0.75 Hz to 0.0024 at 12.0 Hz while at the southern part, it decreases from 0.0089 to 0.0023 at 12.0 Hz. Scattering attenuation has a maximum value at 0.75 Hz as 0.0103 for the northern part and 0.0007 at 12.0 Hz, whereas for the southern part it varies between 0.0096 at 0.75 Hz and 0.0006 at 12.0 Hz. As a sum of two types of attenuation values, total attenuation changes at the northern part from 0.0179 at

0.75 Hz and 0.0025 at 12.0 Hz, while for the southern part it is 0.0169 at 0.75 Hz and 0.0024 at 12.0 Hz. At each frequency, attenuation values are relatively higher in the north than in the south of the CAFZ.



**Figure 5.** Fitting example of an earthquake shown as red star recorded at four stations as black triangles. Surrounding main cities are marked by black squares. Bathymetry and topography data are taken from GEBCO (2019), and active faults are from Şaroğlu et al. (1992). Beneath the topography map, the five columns represent the central frequencies and the rows corresponding to station recordings. Red and blue colors show the synthetic, and the observed energy envelopes. Grey indicates the smoothed observed energy densities. Dots show the average of respective energy densities in the S-wave window.



**Figure 6.** Topography maps of the study area showing also 2-D variation of the inverse quality factors $(Q_I^{-1}, Q_S^{-1}, Q_T^{-1})$ which represents seismic intrinsic (left), scattering (middle) and total attenuation (right), respectively. Each row corresponds to the investigated frequency bands increasing from top to bottom. Attenuation value increase is color-coded and increases from blueish to reddish colors. Stations are shown as triangles. The dashed line corresponds to the Central Anatolian Fault (CAF) and other major faults in the study area indicating the tectonics. Black stars mark the volcanic provinces. Annotations are the same as in Fig. 2. ATB, AB and KB refers to Anatolide-Tauride Block, Adana Basin and Kırşehir Block.





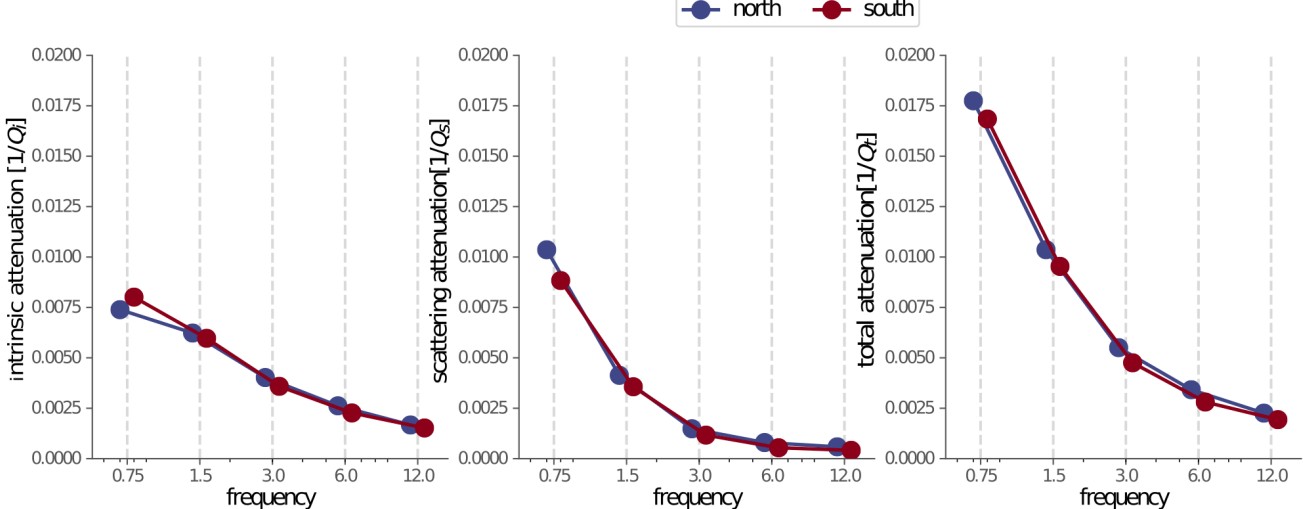

**Figure 7.** From left to right inverse of the quality factors $(Q_I^{-1}, Q_S^{-1}, Q_T^{-1})$ as a function of frequency. Where northern part is shown with blue and southern with red.

## 4 Discussion and Conclusions

Our approach employing a joint inversion technique to estimate the structural properties provides powerful tools to investigate the crustal structure of tectonically active regions. Izgi et al. (2020) reported a clear significance of the scattering attenuation over the intrinsic one at the frequencies corresponding to the deeper parts of the upper crust as well as an overall decrease in attenuation properties towards the north of the western NAFZ. Mapping the variation of attenuation properties enabled a clear marking of the main branch of the central part of the CAFZ as this was the case in early attenuation modeling studies using the same approach at the western and central part of the NAFZ (Gaebler et al., 2019; Izgi et al., 2020). Similarly, we could define the TGFZ by a considerable contrast in attenuation variation (Fig. 6). The volcanic provinces in the Central Anatolia region show the highest attenuation. When examining the 2D spatial distributions of attenuation values, two main blocks are clearly distinguished: the Kırşehir block and the Anatolide-Tauride block that are separated by the CAFZ. The younger and hotter crust exhibits dominancy of intrinsic attenuation (Eulenfeld and Wegler, 2017) since the anelastic processes are controlled mainly by the temperature (Cormier et al., 2011). Ates et al. (2005) reported high heat flow values (around 70-80 m) in the Central Pontides, resulting in shallow Curie temperature depths and the presence of hot materials. Intrinsic attenuation is found to be slightly dominant over scattering attenuation within the region except at the deepest part (0.75 Hz) as this can explain the presence of thick volcanic rocks beneath Central Anatolia. Significant differences from the northern to the southern parts of the CAFZ imply a highly heterogeneous nature of the region with its complex tectonic structure. Part of the observed contrast can be denoted to lithological contrasts that are likely developed due to the structural–tectonic relationships in the crust. These



contrasts have a potential impact on the regions of distinct sedimentary and crustal architecture. To give a better overview of
our findings, we examine our results in three sections: Kırşehir Block, southeastern and southwestern parts of CAFZ.

## 4.1 Southeastern part of the CAF

The southeastern region of the CAFZ comprises the Central Tauride Mountains, which are bounded by the Kozan Fault (KF) to
the south and the Adana Basin (AB) and Sarız Fault (SRZ) to the east. Faccenna et al. (2006) investigated Pn velocities in this
area and interpreted the sharp transition between fast and slow velocities as the boundary of a second slab window extending
towards the west. We observe a gradual decrease in attenuation values from the eastern edge of the study area including the
Central Tauride Mountains to the southeastern part of CAFZ which is consistent with relatively high P- and S-wave speeds
beneath the Mountain range, resolved in early studies (Gans et al., 2009; Abgarmi et al., 2017; Wang et al., 2020; Eken et al.,
2021). Significant crustal thinning (Eken et al., 2021) is evident in the transition from the Taurus Mountains (more than 40 km
thick) to the Adana Basin (about 30 km thick) over a relatively short lateral distance of about 60 km. This pronounced thinning
of the crust corresponds to the Kozan fault, which is thought to be a transtensional splay fault branching from the East Anatolian
Fault Zone (EAFZ) (Higgins et al., 2015). However, the observed motion along the Kozan fault is insufficient to account for
the 2 km of uplift of the Taurus Mountains since the Late Miocene. The uplift of the Taurus Mountains and Central Anatolia
has been a subject of interest and debate in the scientific community. This uplift coincides with a decrease in the attenuation
values of within this area and the Kozan Fault is characterized by the marked change in scattering attenuation from low to
high in deeper parts and from intermediate to lower in shallower parts. The Tauride Mountains, located at the southern border
of the Anatolian microplate, are a part of the Anatolide Tauride block. These mountains are composed of nonmetamorphosed
platform carbonates and are covered by late Miocene platform carbonates. Two recent competing hypotheses, the slab break-
off model (Schildgen et al., 2012, 2014) and drip tectonics (Göğüş et al., 2017) attempt to explain the uplift of the Taurus
Mountains and Central Anatolian Plateau. Recent constraints on the Moho topography Eken et al. (2021) indicated that the
relatively thick crust in the Taurus Mountains aligns with the slab break-off model's assumption of simple crustal isostatic
equilibrium. However, the crustal thickness model is only similar to the models of lithospheric removal in the plate hinterland
of Central Anatolia. Relatively high velocity anomalies resolved previously in Wang et al. (2020); Confal et al. (2020), and
low seismic attenuation properties in this work for the crust and mantle underneath the Taurus Mountains provide further
evidence for the presence of lithospheric remnants in the sublithospheric mantle. Cosentino et al. (2012) claimed that the uplift
in the area is the possible outcome of the asthenospheric upwelling following the slab breakoff. Their model has required
hot material in the crust-mantle volume in the north of the Taurus Mountains. This was found to be consistent with profound
low P-wave velocities (1–2%) at shallow depths but a large-scale low velocity (2–4%) zone revealed in the lower crust and
uppermost mantle as observed in a high-resolution 3D seismic tomography study conducted by Wang et al. (2020) beneath the
Late Cenozoic Volcanic Centers in central Türkiye. Adana basin experiences gradual crustal thickening towards the northeast
(Abgarmi et al., 2017; Higgins et al., 2015; Delph et al., 2017; Fichtner et al., 2013a, b), which is evident from a gradual change
in attenuation values. Continuing to the east, the SRZ accommodates internal deformation of the Taurus Mountains (Kaymakci



et al., 2010). This fault exhibits a distinct change in attenuation values, particularly at shallower depths (e.g., 3.0, 6.0, and 12.0 Hz), which decrease rapidly from west to east of the SRZ.

## 4.2 Kırşehir Block

The triangular shaped Kırşehir block, that is bounded by the strand of CAFZ and ITS, is known as Crystalline Complex and mainly composed of high temperature metamorphic rocks with igneous intrusions (Okay and Tüysüz, 1999; Whitney and Hamilton, 2004; Gürer et al., 2016). Localities with pronounced thin crust is distributed mainly along the central Kırşehir block, where the Moho is typically below 35 km and as shallow as $\sim 25km$ (Vanacore et al., 2013; Eken et al., 2021). The faults (Tuz Gölü Fault, CAFZ) and sutures (Ankara-Erzincan suture, Inner-Tauride suture, Izmir-Ankara Suture) by which the

Kırşehir block is bounded seem to accommodate crustal thickness variations between the inside and outside of this block. The close relationship between Moho depth changes, fault and suture zones suggests a strong control of structural inheritance on the present-day crustal structure and fault development. This complex deformation area is well observable via the 2D spatial distribution of our attenuation estimates. There is a noticeable variation in the crustal composition across the CAFZ. For instance, the CAFZ appears to act as a separator between carbonate nappes from the highly deformed metamorphosed rocks and

exploits the lithospheric scale weakness of the ITS (Fig. 2). On the northwestern side of the zone, the Central Anatolian Volcanic Province (CAVP) consists of a volcanic complex characterized by calc-alkaline to Holocone alkaline volcanic activity during the middle-late Miocene period, following a northeast to southwest direction (Toprak and Göncöoğlu, 1993; Toprak, 1998; Piper et al., 2002; Aydin et al., 2014). The CAVP encompasses pyroclastic lava flow deposits that exhibit an age progression from southwest to northeast (Schleiffarth et al., 2015). The highly attenuating behavior of the southwestern region of the

CAVP can be observed in Fig. 6, in particular, for the deeper sections of the crust (e.g. 0.75, 1.5, and 3.0 Hz). Early seismic imaging studies based on the depth migrated and stacked converted wave analyses (Abgarmi et al., 2017), local P-wave/Pn wave tomography (Gans et al., 2009; Wang et al., 2020), ambient noise tomography (Delph et al., 2015), and modelling of teleseismic P-coda autocorrelation functions (Eken et al., 2021) suggest slow seismic wave speeds beneath most of the CAVP. As evidence for this, observed multiple large negative amplitude conversions (Abgarmi et al., 2017), or a distinct transition

between faster and slower in terms of $P_n$ velocities from the southern to northern part of CAF (Gans et al., 2009) can be given. Our high attenuation estimates and large negative amplitude with moderate Vp/Vs ratio (i.e. 1.75-1.80) (Eken et al., 2021) can be explained by a possible partial melt in this area. High and low electrical resistivity anomalies resolved from the inversion of magnetotelluric data in southwest Cappadocia of Central Anatolia evidenced the covering layer of welded ignimbrites and Quaternary volcanism products as well as a relatively deep ( 4–6 km), single magmatic chamber beneath

Mt. Hasan and the surrounding system (Tank and Karaş, 2020). Similarly, in another magnetotelluric modeling study (Başokur et al., 2022) detected deep and large zone of low resistivity $< 10\Omega m$ interpreted as a shallow-seated magma reservoir at a depth of 3 to 7 km beneath the volcanic edifice of Karadağ that is one of the inactive stratovolcanoes from the Quaternary period in the area. At relatively shallow depths ($> 1 - 2km$), a near-surface localized low resistivity zone of lacustrine sediment was reported also by Başokur et al. (2022). This zone is well marked by our moderate attenuation values at high frequencies (12Hz).

Zones of elevated fluid content exhibit higher attenuation in the Kırşehir block compared to the southern part of the North-



South boundary (see Fig. 7). The fluid-rich content is visible in all frequency bands and characterized by dominant intrinsic attenuation compared to scattering one except 0.75 Hz. The Tuz Gölü Basin is an extensive low-relief region between Pontide and Tauride Mauntains with a basin sedimentation controlled by the extensional Tuz Gölü Fault (Cemen, 1999; Dirik and Erol, 2003; Oezsayin and Dirik, 2007). Biryol et al. (2011) reported upwelling of asthenospheric material from seismic tomography below Tuz Gölü through lithospheric slab breaks and tears. Cosentino et al. (2012) observed a relation of slab breaks to uplift patterns towards the south. We observed a noticeable alteration in both scattering and intrinsic attenuation, particularly in the deeper sections of the crust (e.g., 0.75 and 1.5 Hz), which characterizes the SAVF and the TGFZ. The spatial variation of scattering attenuation, which demonstrates an increasing attenuation towards the south, facilitates the propagation of the slab tear. At a depth of 75 km beneath Central Anatolia which coincides with Kırşehir Massif, a structure showing slow anomalies was detected (Zhu, 2018). High attenuation values in the shallow parts of the lithosphere correspond well with similar findings of Biryol et al. (2011) as well as the results from Confal et al. (2020) which presents low S-wave velocities therefore high $V_p/V_s$ ratios in accordance with intrinsic attenuation.

### 4.3 Southwestern part of the CAF

In this region, young basaltic volcanic rocks are clearly identified by high attenuation characteristics. Specifically, at deeper parts (between 0.75 and 1.5 Hz), this high attenuation aligns well with Neogene volcanism, which originates from the upper mantle and is associated with the presence of Holocene volcanoes in the southwestern part of the CAFZ (Pearce et al., 1990). This correlation proves the existence of recent small-volume mafic volcanic activity and a mafic sill that has stalled in the crust (Rojay et al., 2001; Abgarmi et al., 2017). The study area near the western edge of the CAF demonstrates the highest attenuation values, coherent with findings from a related study by Delph et al. (2017), which revealed the slowest shear velocities with a NE-SW trend in the same region. At higher frequencies (e.g., 3.0, 6.0, and 12.0 Hz), there are observations of elevated scattering and intrinsic attenuation near the ATB. Toward the north of this region, a strong contrast within the lateral variation of attenuation across the CAF appears to confirm the crustal thickening (Gans et al., 2009; Vanacore et al., 2013; Eken et al., 2021) that is considered to develop in response to the mantle upwelling following the detachment of Cyprus slab north of the Tauride Mountains (Melnick et al., 2017). At the lowest frequency ranges (e.g., 0.75, 1.5 Hz), scattering attenuation is notably more pronounced than intrinsic attenuation, particularly in the upper crust. This suggests the potential intrusion of small volumes of mafic sill into the young basaltic rocks in the region. Overall, our findings provide strong evidence of the complex nature of the upper-crustal part of the plateau, which includes volcanic provinces accommodating active deformation due to the relative movements of the Arabian, African, and Eurasian plates. The observed differences in attenuation serve as markers for the aforementioned tectonic and geological features. In short, the decrease in attenuation values in the Southeastern part coincides with the crustal thinning around the Kozan fault which accommodates the 2km uplift of the Central Tauride Mountains. Along the Kırşehir Block, high attenuation characteristics are notably observed in the southwestern part of the CAVP, especially in the deeper sections of the Earth's crust, indicating a possible partial melt. At shallower depths, high attenuation values are associated with a localized low-resistivity zone of lacustrine sediment near the Earth's surface. The presence of fluid-rich material is evident in all frequency bands, primarily characterized by the dominance of intrinsic attenuation, with the exception



of the 0.75 Hz frequency band. The change in both intrinsic and scattering attenuation along the southwestern part of the CAF in deeper sections of the crust marks the SAVF and TGFZ, and increasing scattering attenuation towards the south shows the slab tear propagation.

*Code availability.* The python code (Qopen) used for carrying out the inverse modeling is available under the permissive MIT license and is distributed at https://github.com/trichter/qopen.

*Data availability.* IRIS Data Services, and in particular the IRIS Data Management Center facilities, were used to access the seismic waveforms, associated metadata and/or derived products used in this study.

*Author contributions.* The paper was initially prepared by GI and TE. PG reviewed the entire manuscript, particularly the Methodology section. TK-E reviewed the entire manuscript and contributed to the geo-electrical crustal property in the study region and geo-dynamic implications. TT equally conceived the study and contributed to the paper writing and interpretation of the results and tectonic background.

*Competing interests.* The contact author has declared that none of the authors has any competing interests.

*Acknowledgements.* Gizem Izgi acknowledges support from the University of Potsdam. Tuna Eken and Tuncay Taymaz acknowledge financial support from the Alexander von Humboldt Foundation (AvH) towards computational and peripherals resources. We thank to Christoph Sens-Schönfelder for his valuable comments and helpful discussion and Ceyhun Erman for his contributions to the maps.



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
