# Peer review of "Frequency-Dependent Shear Wave Attenuation across the Central Anatolia Region, Türkiye"

_EGUsphere, 2023_

## Author Response (AR1)

**Reply to Reviewer# 1:**

This is interesting research that expands upon previous studies in the area. Figures are well presented with informative captions, but the main text needs some additions/work.

**Reply:** We sincerely appreciate Reviewer's thorough examination of our manuscript and the valuable insights provided. We have diligently incorporated the suggested changes to enhance the overall quality and coherence of the paper. Changes related to the comments of Reviewer# 1 are marked by blue colors in the manuscript. We believe that these revisions offer a distinct and effective response to the reviewer's feedback, leading to a more refined and impactful manuscript. We deeply appreciate the thoughtful critique and believe these enhancements contribute significantly to the overall strength of our research paper.

There is very little discussion in the uncertainty that may be present in the results. Please present estimations on the error in the attenuation values that may be caused by envelope misfits. Stating "The overall quality of the fits is further comparable to those estimated for different tectonic settings in previous studies" is insufficient. The maps in figure 6 do give some confidence that this error is small as results correlate with known geological features. I can suggest adding errorbars to figure 7 as that may strengthen the intrinsic vs. scattering dominance hypothesis.

**Reply:** Acknowledging the need for a more detailed discussion on uncertainties, we have included a section in the revised manuscript specifically addressing potential sources of uncertainty, with a focus on envelope misfits (please see lines#100-135). Additionally, we have provided error estimates for our attenuation values resulting from these misfits. The incorporation of error bars has been implemented into Fig.7 to visually represent and emphasize potential uncertainties.

The study is also weakened by small mistakes in the text. Important words key to the context of a sentence are occasionally missing and distract from the results of the research. In addition, some paragraphs are as long as a page and would benefit from being broken up a bit to help the reader. The same is with some sentences that are overly long but this is minor and likely a personal preference. I can recommend asking someone who has not read it before to give it a quick look before resubmission.

**Reply:** We have carefully reviewed and revised the main text to rectify instances where important words were missing, ensuring a more coherent context for each sentence. Paragraphs have been strategically divided into smaller sections to enhance readability, and sentence length has been adjusted for clarity.

I don't like how the discussion and conclusions section is structured. For a start, the conclusions are at the beginning of the section, when they should be at the end. Currently, the article ends quite abruptly on a discussion point.

Better highlighting the points relevant to the key outcomes of this study (i.e., presence of partial melt), would also make the article stronger. Section 4.1 seems particularly weak with much of the discussion focusing on other research rather than the additions this study brings. For example, it is hard to understand which of the two competing hypotheses for the uplift of the Taurus Mountains this article is supporting. Sections 4.2 and 4.3 do not have this issue where the impact of the study is well discussed

**Reply:** We appreciate the reviewer for bringing up this aspect. In response to the feedback, we have reorganized the Discussion and Conclusion section in the revised version. Conclusions now appear at the end of the manuscript as a new section, creating a more conclusive and coherent ending to the manuscript. We believe that the restructured Discussion section now better emphasizes key messages inferred from the outcomes of this work. Section 4.1 has been modified to focus more on the unique contributions of our study. Conclusions have been added to highlight the relevant points, such as partial melt. Interpreting the attenuation results, the presence of a partial melt, especially notable in conjunction with the zones of geo-electrical features implying elevated fluid-rich content in the crust. These zones exhibit a distinctive character marked by high and dominant intrinsic attenuation, indicating a correlation with the geological processes associated with partial melting and the presence of significant fluid content. Moving southeast, a noteworthy pattern is observed in the variation of attenuation parameters, particularly in connection with the Central Tauride Mountains. The observed gradual decrease in attenuation is linked to the geological evolution in this region. Specifically, it is associated with the high altitude of the Central Tauride Mountains that is proposed to be uplifted in response to slab break-off and subsequent mantle upwelling processes. These attenuation trends not only offer valuable insights into the geological

complexities of the region but also contribute to new constraints on how seismic signals are influenced by the dynamic interplay of geological processes within Central Anatolia.

Perhaps restructuring according to geological feature (i.e., Kırşehir Block, partial melting) rather than by area/region would better present the outcomes of the study?

**Reply:** When dividing the discussion section into sub-groups for the original manuscript, we intentionally focused on geographical regions showing distinct properties. Thus, we would like to keep this form of the discussion section. However, in the revised section, we swapped the first and second subsections in order to enhance the flow of the story and ensure proper logic between the sections/subsections.. Several minor revisions have been further made throughout the manuscript to improve overall clarity and presentation.

**Reply to Reviewer# 2:**

Overview and general recommendation:

The study aims to investigate the Central Anatolian Plateau gaining insight into the seismic attenuation properties of the crust. The authors model intrinsic and scattering attenuation separately to provide evidence of the possible source for the geodynamic processes yielding the observed deformation. They perform coda wave modelling by fitting observed and synthetic coda wave envelopes for each earthquake in multiple frequency bands. Radiative transfer theory was utilized for the forward modelling of the synthetic coda-wave envelopes of local earthquakes. The contribution of this study to the target area is interesting. The authors clearly introduce the regional deformation and tectonic features and the used dataset. Even though the method and findings are discussed in detail, some additions could improve and strengthen the manuscript. Furthermore, the figures need some improvements.

**Reply:** We would like to express our gratitude to Reviewer 2 for the thoughtful and constructive feedback on our manuscript, which has undoubtedly strengthened the robustness of our study. We have carefully addressed each of the major comments and suggestions, incorporating necessary changes to improve the clarity and completeness of the manuscript. Changes related to the comments of Reviewer# 2 are marked by red colors in the manuscript. We believe these revisions enhance the overall quality and clarity of the manuscript.

Major Comments:

- Even though there are references for the inversion process, when the authors describe the minimization problem (line 110), I suggest briefly explaining how the optimized g* is obtained and how the minimization process works (i.e. which are the main steps). Equations 3 and 6 should be introduced (check the logarithm in Eq 6).

   **Reply:** We have reformulated this section. The revised section has an improved flow now explaining synthetic and observed calculations of coda envelopes and how they are compared within the inversion process within their basic theoretical frames. Finally, we provide a simple stepwise procedure to indicate how g* and b are optimized.

- Section 2.2: the explanation of the fit and the accuracy should be improved (e.g., what they mean by 'smaller fit' is not clear). The selection of the S- and coda windows for each seismogram should be better stated (also in lines 75-79).

   **Reply:** We have explained the selection of S wave and coda wave windows by introducing the following sentences to lines 85-91 of the revised manuscript instead of the brief explanation. Also, the 'smaller fit' annotation is changed, and the sentence became more explanatory as in the current version being the following: 'Thus, the bottom row in Fig. 5 representing the distance equal to 105 km, shows a smaller portion of the observed envelopes fit to the synthetic envelopes.'

- When introducing results and also the study area, the figures have to be mentioned. The results paragraph begins with no references to the figure. Besides, the reference to figures and panels is often missing (e.g. Lines 136-137, 156-161, 167-168, 200). The reference to the maps will

enhance the discussion by making clear what the findings of this study are, in comparison to previous works.

**Reply:** We appreciate the Reviewer# 2 for this suggestion. We agree with this point. In the revised manuscript we have made the requested changes by inserting reference figures to the results accordingly.
- Line 180: are the authors stating that frequency-dependence provides depth variations? This is mentioned only later on (line 197).

**Reply:** This study does not provide a concrete depth variation of attenuation parameters but intend to correlate with existing geophysical/geological constraints from other studies to make approximate depth interpretation.
- The authors discuss the findings in comparison with previous results. However, in some parts, the correlation between the attenuation results and other geophysical observations could be made clearer. Besides, I suggest that a concluding section separately from the discussion would be helpful to stress the results and the contribution of this study.

**Reply:** We appreciate the Reviewer# 2 for this suggestion. This was the concern from the Reviewer# 1 too. We have separated the Discussion and Conclusion sections accordingly. We believe that the restructured Discussion section now better emphasizes key messages inferred from the outcomes of this work. Conclusions now appear at the end of the manuscript as a new section, creating a more conclusive and coherent ending to the manuscript.

The authors have to check the wording of the sentences and citations (e.g. lines 25, 28, 62, 90, 91, 184, 187).

**Reply:** We thank the Reviewer#2 for pointing out the errors between these lines. We have rephrased the sentences to correct grammatical errors and ensure adherence to the citation style.

Figures:

- The authors should label the CAFZ in Figure 1. In the caption, it has been said that the extensional boundaries are shown in green.

**Reply:** Thank you for this correction. CAFZ is now marked by green in Figure 1.
- Figure 2: magenta polygons are not clear. The authors could try to use an edge and a marker of different colors. If correct, the authors want to highlight the plateau-like morphology. However, the used colormap is not good for visualization and also three different colormaps are used in three different figures (1-2-4). Why don't they use the same colormap as in Figure 4? What are the magenta dashes at the edge of the figure? The authors should label the Anatolide-Tauride block that is described in the text.

**Reply:** Figure 2 in the original manuscript was adapted and modified from the figure in Abgarmi et al. (2017). In the revised version we provide a more detailed and recent topographical information. We further decreased the dominancy of the topography to highlight the plateau-like morphology. We further changed magenta colors to red, and labeled the Anatolide-Tauride block. Concerning Figure 1 and 4, thank you for your suggestion. Now we use the same colormaps for these maps.

Figure 3: Using a lighter blue and red could help the visualization. It would be also helpful if the authors found a good way to add CAF and ITS labels (which are mentioned in the caption) to the map.

**Reply:** We appreciate the Reviewer#2 for this suggestion. Lighter colors now are used and the labels are implemented in the revised version of the manuscript.
- Figure 4: the right edge of the figure has been cut off. The colorbar can be positioned outside the map. The white triangles can be drawn with a thicker edge to make them more visible.

**Reply:** Figure 4 is improved according to the reviewer's comments. We used thicker edge colors for marking the stations, changed the position of the colorbar and completed the right edge of the figure.
- I suggest to work also on the labels (in all the figures) as some of them are not clearly legible.

**Reply:** Required changes at the request of reviewer #2 have been made accordingly.

  •Figure 6: if the colorbar is outside, it is more visible. Why aren't the stations shown in black? Currently, they are not distinguishable from the background. All the labels should be on the same map (the labels can be repeated whenever an attenuation contrast, which is correlated to a specific feature, is found in the maps), otherwise, it may be difficult to follow the discussion in the text.

**Reply:** We placed the colorbars outside the figures as suggested, so they became more visible. We do not show stations in black not to block the visible attenuation values at some the stations. To increase the visibility of station locations and maintain the background attenuation values still notable, however, we implement black edge colors for the triangles. In the revised version we present all the labels on the same map (i.e. 0.75Hz-Total Attenuation map). However, we still keep some specific labels at their original positions. This is because they correspond to specific regions with characteristic attenuation values of a different frequency band interpreted in Discussion section.

---

## Author Response (AR2)

We thank to the Reviewer 2 for the detailed inspection and comments which improved the quality of the manuscript. Two figures as mentioned are revised by editing the labels of the Figure 6 and in Figure 3 cropped part of the figure is improved now. Also, lnEobs=lnEmod is used to minimize the error function which is calculated using least squares method mentioned now in the text in more detail.

We thank additionally to Michal Malinowski (Topic Editor) and Susanne Buiter (Executive Editor) for their valuable comments and their quick responses.